# Decoupling the origins of irreversible coulombic efficiency in anode-free lithium metal batteries

Chen-Jui Huang [1], Balamurugan Thirumalraj [1], Hsien-Chu Tao[1], Kassie Nigus Shitaw [1], Hogiartha Sutiono[1], Tesfaye Teka Hagos[2], Tamene Tadesse Beyene[1], Li-Ming Kuo[1], Chun-Chieh Wang[3], She-Huang Wu[2], Wei-Nien Su [2] & Bing Joe Hwang [1,3,4 ✉]

Anode-free lithium metal batteries are the most promising candidate to outperform lithium metal batteries due to higher energy density and reduced safety hazards with the absence of metallic lithium anode during initial cell fabrication. In general, researchers report capacity retention, reversible capacity, or rate capability of the cells to study the electrochemical performance of anode-free lithium metal batteries. However, evaluating the behavior of batteries from limited aspects may easily overlook other information hidden deep inside the meretricious results or even lead to misguided data interpretation. In this work, we present an integrated protocol combining different types of cell configuration to determine various sources of irreversible coulombic efficiency in anode-free lithium metal cells. The decrypted information from the protocol provides an insightful understanding of the behaviors of LMBs and AFLMBs, which promotes their development for practical applications.

[1] Department of Chemical Engineering, National Taiwan University of Science and Technology, Taipei, Taiwan. [2] Graduate Institute of Applied Science and Technology, National Taiwan University of Science and Technology, Taipei, Taiwan. [3] National Synchrotron Radiation Research Center (NSRRC), Hsinchu, Taiwan. [4] Sustainable Energy Development Center, National Taiwan University of Science and Technology, Taipei, Taiwan. ✉email: bjh@mail.ntust.edu.tw

Lithium metal, with an ultrahigh theoretical specific capacity (3860 mAh g$^{-1}$) and low redox potential ($-3.040$ V vs. standard hydrogen electrode), has already been extensively investigated over the four decades[1,2]. However, lithium metal batteries (LMB) still suffer from several barriers and yet to be commercialized. More specifically, the safety issues induced by Li dendrite growth and internal short circuit (ISC)[3], poor efficiency attributed to the formation of high surface area lithium (HSAL, dendrite) and dead Li[4,5], and severe electrolyte decomposition at the negative electrode leading to electrolyte dry-up and the formation of thick solid electrolyte interphase (SEI) that increases the internal resistance and consumes the electrolytes[6–11].

To overcome the aforementioned challenges, one has to systematically study Li metal stability/protection[12], SEI formation mechanism[13,14], and suppression of Li dendrite growth in LMB[15–17]. Many works have also been done by using different electrolyte formulas[18], 3D architecture Li[19], and artificial coating layers[20] to study their effect on increasing the electrochemical performance of LMB. Meanwhile, several key factors would still affect the cycling performance of LMB and are crucial in achieving high specific energy of 500 Wh kg$^{-1}$ demanded by electric vehicle energy-storage market such as electrolyte amount[21], temperature[22], pressure[23], amount of Li or cathode[24,25], and current density applied[9], etc. Recently, anode-free lithium metal batteries (AFLMBs) are considered as phenomenal energy-storage systems owing to higher energy density than that of LMB, which with excess Li in the system, and greatly reduced safety risks since no Li metal is used during cell manufacturing, which remarkably increases the simplicity of cell fabrication and reduces the cost of cell assembly, too[26–31].

However, in most of the published works, the electrochemical performance of LMBs/AFLMBs is often discussed by comparing capacity retention, reversible capacity, or rate capability, which easily overlooks or even misunderstands the information that is concealed by the meretricious results when adopting only one or two points of view. To systematically evaluate the electrochemical performance of both LMBs and AFLMBs, and unfold all the messages hidden within the battery, one has to comprehensively examine the information from all the possible perspectives. More importantly, integrating all the unraveled phenomena and messages to have a better overall evaluation of the battery systems is essential. One efficient way is to study the irreversible coulombic efficiency (irr-CE), which may represent the side reactions and sources of capacity loss in the battery. Meng et al.[32] demonstrated an analytical method of titration gas chromatography (TGC) to quantify the contribution of dead Li to the total irr-CE in Li/Cu cells, identifying dead Li as the major reason accounted for the capacity loss of Li/Cu cells. They comprehensively discussed the formation of inactive Li and determined the origin of irr-CE of Li anodes within Li/Cu cells by decoupling the dead Li and SEI formation, which provides strategies for more-efficient Li plating and stripping on Cu substrate. However, the important information of the irr-CE or capacity loss from the cathode and cross-talk effects in cathode/Li or anode-free cathode/Cu cells could not be extracted from the TGC method. Thus, there is still a lack of holistic methodology to identify and quantify the irr-CEs in LMBs and AFLMBs.

In this work, we systematically study four different types of cell configuration, including Li/Li symmetric cells[33], Li/Cu cells, cathode/Li cells, and cathode/Cu anode-free cells, as an integrated protocol to unfold the intrinsic reasons and contributions of individual irr-CEs from not only Li anodes in Li/Cu cells, but also the cathodes in cathode/Li half-cells. Furthermore, the cause of low irr-CE of AFLMBs can also be determined at different states of the batteries through the proposed protocol. Meanwhile, we also observed dendritic Li induced ISC and visualize the formation of dead Li in a Li/Cu cell using in situ optical microscopy (OM) and transmission X-ray microscopy (TXM), and proposed the mechanism of Li nucleation and deposition/dissolution on Cu. This work provides an overall understanding and quantification to the irr-CEs from the full spectrum of different cell configurations such as first extra SEI, dead Li and subsequent SEI, cross-talk effects, first cycle intrinsic irreversible capacity of the cathode, and subsequent oxidative electrolyte decomposition. The proposed protocol could serve as a platform from an overall perspective to evaluate the performance of LMBs and AFLMBs and can be widely applied to various systems for the development of next-generation high-energy batteries.

## Results

**In situ OM observation of Li deposition/dissolution**. Li deposition/dissolution upon cycling with the subsequent dendrite growth and formation of dead Li are observed by in situ OM in Fig. 1a (Supplementary Movie 1, 2, and 3). In the beginning, both

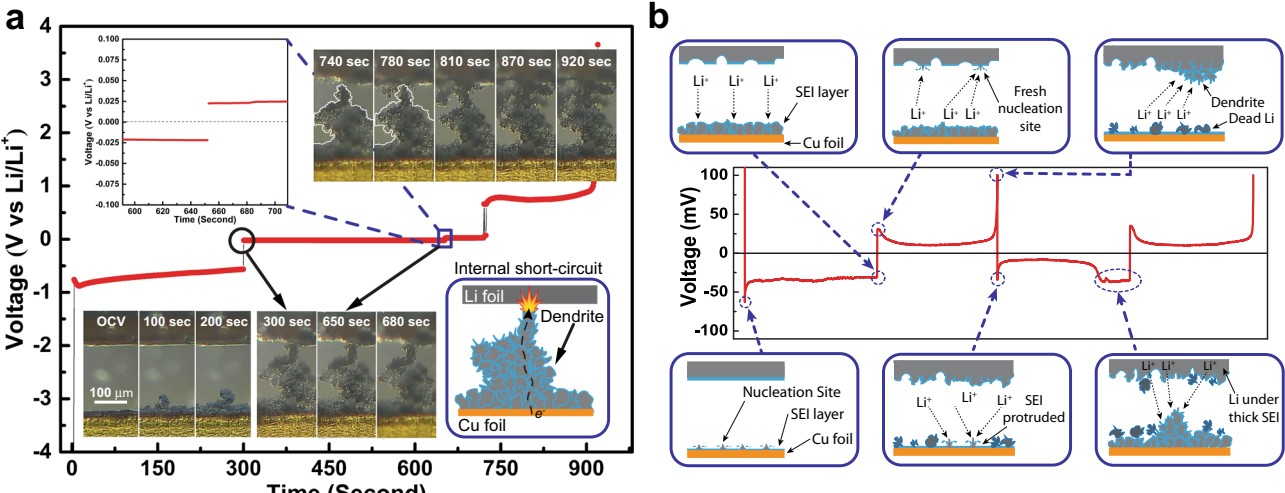

**Fig. 1 In situ OM and Li deposition/dissolution scheme. a** In situ OM measurement of Li deposition/dissolution on Cu electrode under an ultrahigh current density of 500 mA cm$^{-2}$ for demonstration of short-circuiting. It should be noted that the observed short circuit region is different from the initial observation region for Li-plating process. **b** Scheme of Li deposition/dissolution on Cu foil during cycling of Li/Cu cell under a current density of 0.2 mA cm$^{-2}$.

Cu and Li surfaces are very smooth. In the first 10 sec of plating, there was a clear potential drop observed from the curve, which was owing to the overpotential induced by the initial nucleation of Li on Cu foil and SEI fracture. After the nucleation sites were formed, the polarization was found reduced. The plated Li was homogeneous and dense during the first 100 sec; however, granular Li started to grow on different spots of Cu foil owing to the inhomogeneous plating of Li. After the initial granular Li emerged, which served as a fresh nucleus, HSAL vigorously grew on top of the granule rather than the compact and uniform Li beside it on Cu foil. In addition, non-uniform dissolution of Li and formation of a rough surface at the Li electrode were also observed during the first 300 sec, induced from the locally heterogeneous current distribution, which also accelerated the formation of dendritic Li on the Cu electrode (Supplementary Movie 1).

As the plating process continued, the HSAL eventually contacted the Li electrode and short-circuits occurred after 300 sec of Li plating. Then, the cell voltage was found suddenly lifted to −0.022 V vs. Li/Li$^+$ but not zero, indicating there is an SEI resistance between two contacted electrodes. Although the potentiostat was still applying negative current, no extra Li was plated on the working electrode during the short circuit. When the cell was switched to a stripping mode at 650 sec, the slightly positive cell voltage of 0.022 V vs. Li/Li$^+$ was also observed in the stripping process owing to the SEI resistance between both contacted electrodes as in the deposition process. Besides, the morphology of the dendrite was kept unchanged without Li deposition/dissolution during the short circuit (Supplementary Movie 2).

After 740 sec at the stripping process, the short-circuited HSAL between two electrodes suddenly disconnected and started to dissolve. However, dead Li was discovered after 40 sec of stripping as the HSAL stopped dissolving owing to the higher charge resistance, leaving a large amount of dead Li on the Cu surface at the end of the stripping process. This results in poor CE observed from the cycling performance of the cell (Supplementary Movie 3).

**Li nucleation and deposition/dissolution mechanism**. Based on the OM (Fig. 1a) and TXM (Supplementary Fig. 1, Supplementary Movie 4 & 5) observation of Li plating/stripping, the electrochemical phenomena of Li plating/stripping on Cu are investigated, and the corresponding proposed mechanism is shown in Fig. 1b. During the very beginning of Li deposition on Cu foil, an initial energy barrier needs to be overcome by first forming nucleation sites on the clean Cu surface, which causes the initial higher overpotential seen from the curve. Then, Li underneath the thick SEI layer on Li foil starts to dissolve and deposit onto the nucleation sites on the Cu surface, accompanied by the decrease of overpotential. Next, in the Li-stripping process from the Cu surface, the overpotential is induced from the resistance of the SEI layer protruding and fresh Li growth on the Li surface. At the end of the Li-stripping process from the Cu surface, the cell voltage suddenly lifted due to the total consumption of active Li, leaving some dead Li at the Cu surface. From our previous in-operando TXM results[4], we observed that the formation of dead Li is generally from the outer part of deposited Li, which disconnects with the active Li at the electrode surface, similar to the observation in this work, and the stacking of dead Li is also observed after several plating/stripping processes. At the same time, on the other side, dendrites start to evolve on the roughened Li surface owing to non-uniform current distribution from the previous dissolution process. In the following cycles, the deposition/dissolution of Li is similar to the first cycle in general, including SEI fracture, fresh Li nucleation, dendrite growth, and dead Li formation at both sides. More detail regarding the SEI fracture mechanism is discussed in our previous

work[34]. However, it is worth noted that in the latter part of Li deposition in the second cycle, a second plateau with the potential similar to the deposition overpotential in the first cycle appeared. It is suggested that the first and second plateau is ascribed to the freshly deposited Li (HSAL) stripping and the bulk (original) Li, which is covered by the thick SEI owing to the low reduction potential of metallic Li, stripping on the Li side, respectively. Thus, the overpotential relates to Li deposition/dissolution processes on the Li surface is significantly influenced by the nature of SEI, the HSAL on both Cu, Li surfaces, and the charge transfer processes at both interfaces.

**Proposed integrated protocol**. In this section, the four types of cell configuration as an integrated protocol assembled by Li/Li symmetric cell, Li/Cu cell, cathode/Li cell, and cathode/Cu anode-free cell was applied to comprehensively evaluate the performance of LMBs/AFLMBs and unravel the unseen messages concealed in the individual candidates. Figure 2 shows the charge/discharge profiles of all the cell configuration used in the protocol. Figure 3 shows the schematics of each cell configuration at its fully charged and discharged state with the scheme of the proposed protocol from both irr-CE and capacity perspective. By applying the integrated protocol, one can decrypt different information from each cell set-ups and dissect the origins of irr-CE and capacity lost within LMB/AFLMB quantitatively.

Li/Li symmetric cell, with its charge/discharge curves shown in Fig. 2a, is a useful configuration that allows us to understand the kinetics of Li deposition/dissolution by extracting the initial nucleation overpotential and polarization information[12,33,35]. It is also a powerful tool to investigate ISC, critical current density phenomenon, or SEI polarization on Li surface, which although depends on the surface quality of both Li electrodes. However, irr-CE is unable to be obtained from Li/Li cell since the CE is always ~100% due to excess Li from both Li electrodes that compensate the formation of dead Li and active Li lost from reductive electrolyte decomposition (SEI formation or gas generation). As a result, the consumption of active Li owing to the aforementioned reactions is invisible; thus, seldom discussed and often overlooked in this cell configuration[4]. Generally, it is not easy to report reliable and reproducible information on cycling performance or even ISC of this Li/Li protocol unless the quality of Li surface employed, the amounts of electrolytes, and the pressure on the tested cell are well-controlled.

However, by replacing Li with Cu as the working electrode in Li/Cu cell as shown in Fig. 2b, it is possible to quantify the inactive Li on the Cu electrode from the irr-CE of the Li/Cu cell in each cycle. Figures 3a and 3d show the scheme of Li/Cu cell at fully plated and stripped state in the first cycle. In the Li/Cu cell, as the excess Li is from the Li electrode, resulting in the Cu side as the limiting electrode. As there is no excessive metallic Li on Cu as it was in Li/Li cell to compensate for the irreversible consumption of active Li on Cu. The irreversible phenomena observed mainly reflect the behaviors of the Cu electrode. In general, the irr-CE in Li/Cu cells can be separated into two main sources, namely dead Li and SEI formation. In particular, the irr-CE is normally higher in the first cycle of Li/Cu owing to the initial extra SEI formation on the Cu surface, causing a larger irr-CE than that in the subsequent cycles, denoted as the first extra SEI (red bar in Fig. 3g). Thus, we proposed that the irr-CE of Li/Cu cell in the first cycle contains both first extra SEI formation and dead Li+ sub. SEI (green bar in Fig. 3g). We have combined the contribution of dead Li and subsequent SEI owing to the fact that it is not possible to easily separate the fraction of which unless an experimental method like the TGC method is performed, and the fraction of subsequent SEI is not stable in

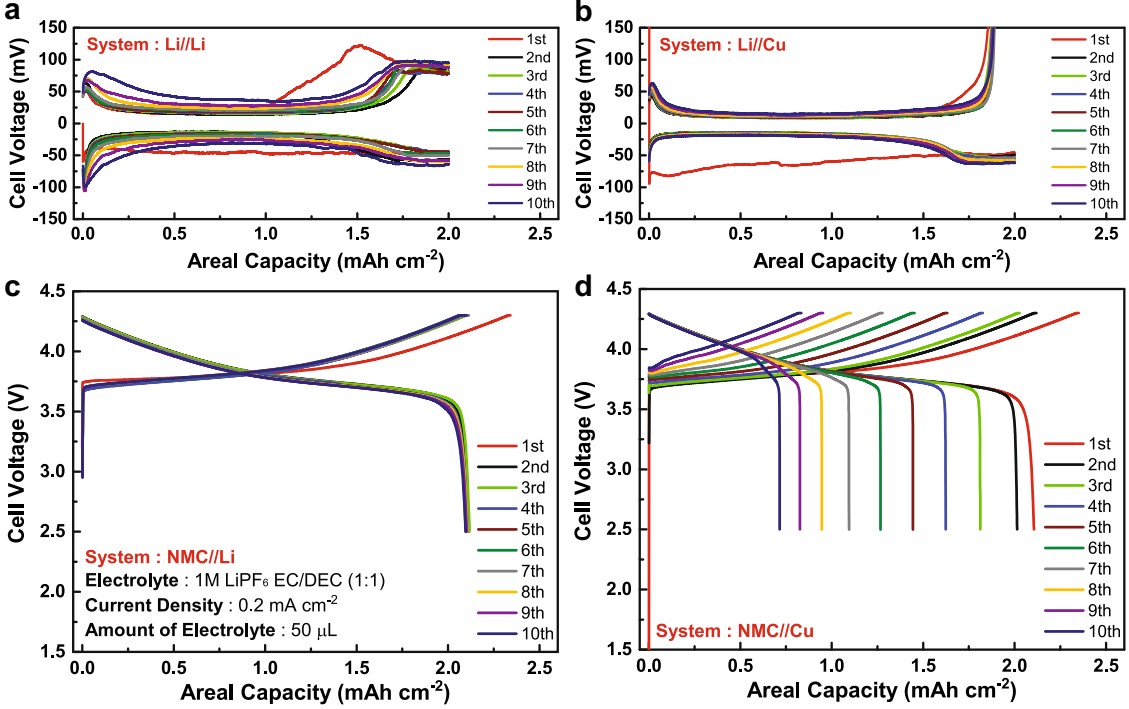

**Fig. 2 Electrochemical charge and discharge curves. a** Li/Li. **b** Li/Cu. **c** NMC/Li. **d** NMC/Cu. All the cells use 1 M LiPF$_6$ in EC:DEC (1:1) as electrolyte with the current density of 0.2 mA cm$^{-2}$. We selected NMC-111 as the cathode material used in this electrolyte. One can substitute the desired cathode material for cathode study in cathode/Li cell and AFLMB.

the subsequent cycles owing to the dendritic Li caused fracture[34]. Therefore, the irr-CE originated from the first SEI formation (red bar in Fig. 3g) can be calculated by subtracting the value in the first cycle with that of the second cycle from the Li/Cu cell protocol. To conclude, the Li/Cu cell is a useful protocol to provide reliable information for the study of electrolyte[27,28,31], and surface engineering approaches[29,36,37] for mitigating the irr-CE ascribed to dead Li and SEI formation.

The third protocol is a cathode/Li cell, and the charge/discharge curves are shown in Fig. 2c, namely a half-cell for studying phenomena taking place at the cathode. When the cathode/Li cell is fully charged, Li$^+$ is de-intercalated from NMC and plated onto the Li anode along with the formation of dendritic or mossy Li (Fig. 3b). In reverse, Li$^+$ is stripped from the Li anode with some dead Li left on it and intercalated back into NMC (Fig. 3e). However, as there is a significant amount of active Li on the Li electrode compared with that in the cathode electrode, i.e., the capacity ratio of the anode to cathode (A/C) is >1 and cathode is the limiting electrode, the excess metallic Li will compensate the active Li loss due to dead Li formation and reductive electrolyte decomposition at the anode side, leading them invisible from the irr-CE observed. As the irreversible reactions at the anode cannot be observed, this protocol acts as an efficient tool to extract information relating to the irreversible reactions at the cathode, including oxidative electrolyte decomposition (Ox. E.D.), cathode degradation, and first intrinsic irreversible capacity of cathode material (first irr-cap. of cathode) in the first cycle[38–42]. Generally, the irr-CE of cathode/Li cells in the first cycle is often found larger due to the initial Ox. E.D. and the correlated cathode-electrolyte interphase (CEI) formation than that in the subsequent cycles. In particular, the first irr-capacity of the cathode is significantly larger and often observed in layered oxide cathode materials, which can be attributed to mainly the slow lithium kinetics at high lithium contents and partially the formation of Li$_2$MO$_2$-like phases. This intrinsic irr-capacity cannot be recovered in the absence of deep

discharge of the cell or unless cycling at the higher temperature to eliminate the kinetics limitation based on our own experiments and the previously reported works (Supplementary Fig. 3)[38–40]. Based on the above mentioned three origins of irr-CE at the cathode, we can dissect the irr-CEs of cathode/Li cell by first quantifying the first cycle irr-CE as the first irr-cap. of the cathode (with Ox. E.D. included, yellow bar in Fig. 3g). Second, in the subsequent cycles, the origins of irr-CEs can be separated into two sources. When the reversible capacity remains the same and stable, the irr-CE of the cell can be attributed to the subsequent oxidative electrolyte decomposition (Sub. Ox. E.D., blue bar in Fig. 3g) with the consequent CEI formation included; however, when the reversible capacity starts to fade, then the irr-CE would become the sum of cathode degradation (denoted as cathode degrad. and shown in gray bar in Fig. 3g) and sub. Ox. E.D. owing to the fact that capacity fading is directly related to cathode degradation. To be more specific, the fraction of cathode degradation can be calculated from the slope of the fitted line of the normalized discharged capacity retention based on equation (1) in the supporting information. Thus, the fraction of sub. Ox. E.D. within the capacity fading region could be finally quantified as the difference between the total irr-CE of cathode/Li cell and that of cathode degradation.

The cathode/Cu cell, which is also called AFLMB as displayed in Fig. 2d, can be recognized as a full cell with the A/C ratio ≅1 since the active Li is purely from the cathode. The two half-reactions taking place in the AFLMB are comparable to those in cathode/Li and Li/Cu cells, namely Li deposition/dissolution on Cu and cathode oxidation/reduction, respectively. When the anode-free cell is fully charged, Li$^+$ is de-intercalated from NMC and plated onto the Cu along with the formation of dendritic or mossy Li (Fig. 3c). In reverse, Li$^+$ is stripped from the anode with some dead Li left on Cu and intercalated back into NMC (Fig. 3f). Therefore, the irreversible capacity and CE of AFLMB are significantly relative to that of Li/Cu and cathode/Li cells, and by

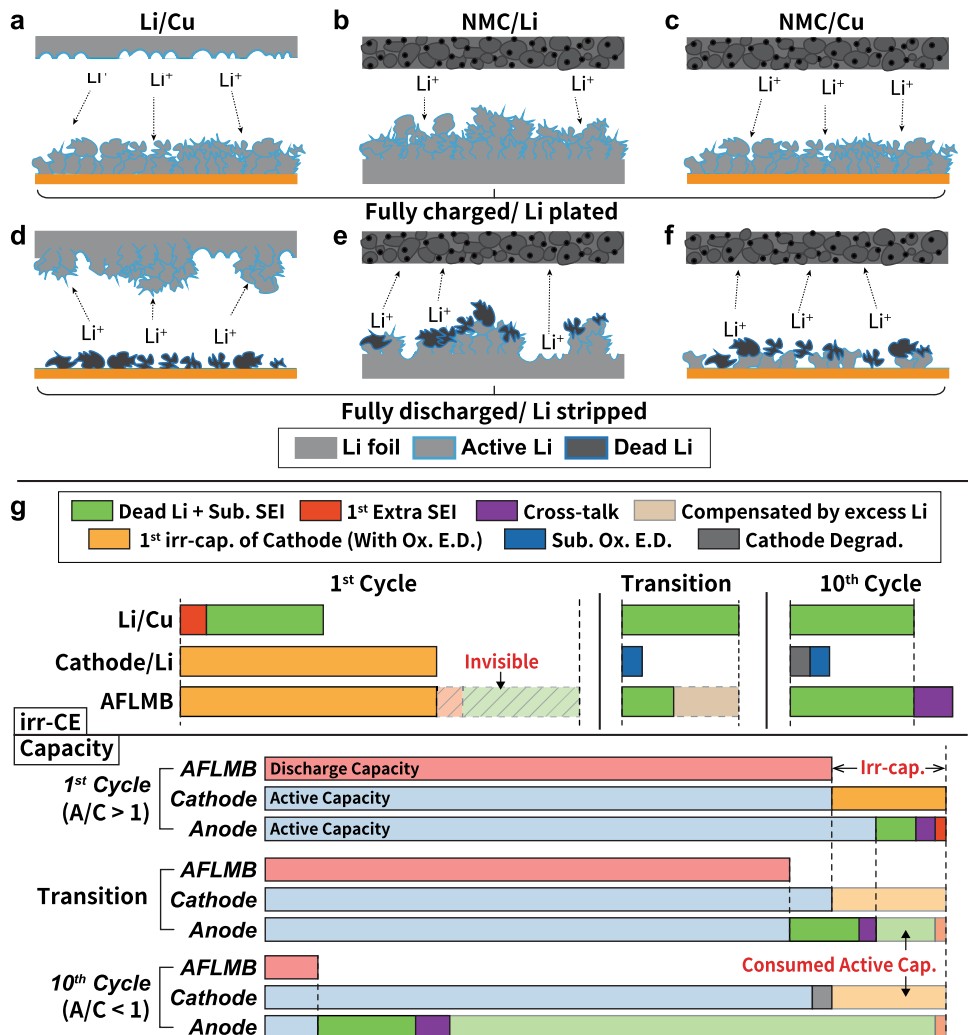

**Fig. 3 Integrated protocol and scheme of different cell configurations at fully charged and discharged states.** Scheme of **a** Li/Cu, **b** NMC/Li, and **c** NMC/Cu cells at fully charged/ Li pated sate in the first cycle, respectively. Scheme of **d** Li/Cu, **e** NMC/Li, and **f** NMC/Cu cells at fully discharge/Li-stripped state in the first cycle, respectively. **g** Proposed integrated protocol to unravel the origins of irreversible CE in AFLMB by Li/Cu and cathode/Li cells. The blue shell on Li represents the SEI layer. Sub. SEI stands for the subsequent SEI formation, Ox. E.D. for oxidative electrolyte decomposition, and cathode degrad. for cathode degradation, respectively.

integrating the information of Li/Cu and cathode/Li cells, it is possible to dissect the proportion of irreversible capacity and CE in the anode-free cell at its different A/C ratio states generated from different origins as shown in Fig. 3g and will be discussed in the following paragraphs. Nevertheless, it should be noted that apart from the sources of irr-CEs and capacity loss in Li/Cu and cathode/Li cells would affect the evaluation of AFLMB; cross-talk effects would also account for the addition irr-CE upon the cycling of AFLMB. During the cycling of AFLMB, cross-talk effects could take place, namely the crossover of transition metal ions from the cathode materials to anode, leading to greatly altered Li plating/stripping chemistry as well as the SEI formation mechanism[43,44]. In other words, more complicated side reactions and higher irr-CE of the Li-plating/stripping processes than those in Li/Cu cell may occur. Therefore, when dissecting the irr-CEs of AFLMB, cross-talk effects should also be considered. The detailed step-by-steps flowchart showing the procedures from identifying the irr-CEs in both Li/Cu and cathode/Li cells and transferring those into AFLMB is provided in the Supplementary information (Supplementary Fig. 3).

As the irr-capacity of anode and cathode could changes along with the cycles in an AFLMB, its A/C ratio and limiting electrode would also change, which would affect the analysis of the contribution of irr-CE. Therefore, it is important to know how the capacity changes in both electrodes. From the perspective of capacity, if the cathode material used possesses the first intrinsic irr-capacity of layered oxide cathode, excess active Li would be left on Cu in the form of metallic Li causing the A/C ratio >1 and cathode as limiting electrode after the first cycle, causing the irr-CE of AFLMB only contributed from the cathode as that in an NMC/Li cell, namely first irr-capacity of cathode and sub. Ox. E.D. In contrast, when the A/C ratio is <1, the irr-capacity of AFLMB would be only ascribed to the anode. To study the effect of A/C ratio on the capacity retention of (AF) LMBs, Supplementary Fig. 6 and Supplementary Fig. 7 show the charge/discharge profiles and cycling performance results of NMC/Li cells with different A/C ratio. When the reversible capacity is kept almost the same as that of the first cycle discharge capacity as if in cathode/Li cell, it can be reckoned as the A/C ratio is >1 where all the irr-capacity observed arouses from the cathode. (Supplementary Fig. 7b) However, the hidden chemistry beneath this phenomenon is the continued consumption of excess Li. This is because dead Li forms in each cycle, yet the active Li inventory within AFLMB cells is solely provided by the cathode and limited. The active Li would be

continuously consumed and eventually used up, as shown in Supplementary Fig. 7a. Thus, as the irr-capacity caused by the cathode degradation is significantly low compared with dead Li formation, the observed reversible (active) capacity of AFLMB can serve as an indicator of the A/C ratio of the cell in situ.

During the continuous consumption of active Li inventory, there will be a transition state in a specific cycle when the A/C ratio is turning into <1. From Supplementary Fig. 7b, this unique circumstance can be characterized by a sudden fading slope transition of the discharge capacity of AFLMB, implying the excess Li is no longer enough to compensate the dead Li formation in each cycle, which leads to less active Li at the anode than that at the cathode, limiting the observed reversible capacity of AFLMB as shown in Fig. 3g. Therefore, the irr-CE observed will combine both anode and cathode in the transition state and dominated by dead Li and subsequent SEI formation afterward. To conclude, the larger the A/C ratio and less severe dead Li formation, the later the transition state occurs as shown in Supplementary Fig. 7b. Thus, by enhancing the CE of Li plating/ stripping and suppressing dead Li formation, the capacity retention and the high CE region can be improved and prolonged in AFLMBs, respectively.

In view of irr-CE, the first cycle irr-CE of AFLMB can be explained by the sum of first SEI formation, dead Li+ sub. SEI, and cross-talk effect when the initial A/C ratio is <1. The fraction of first SEI formation and dead Li+ sub. SEI can be transferred from Li/Cu cell, and by subtracting the value of total irr-CE with that of the aforementioned value in Li/Cu cell, the fraction of cross-talk effects (purple bar in Fig. 3g) can be obtained. As in the subsequent cycles, the irr-CE of AFLMB will be the combination of dead Li+ sub. SEI and cross-talk effects. Meanwhile, when the initial A/C ratio is >1, namely cathode is the limiting electrode, then the first cycle irr-CE is dominated by cathode and equals to the fraction of the first irr-capacity of cathode in cathode/Li cell. In the subsequent cycles, the irr-CE can be separated into cathode degradation and sub. Ox. E.D. when A/C ratio remains larger than one, namely before the transition state. However, the value of cathode degradation in AFLMB may not necessarily equal to that in cathode/Li cell considering the mechanism may be different among two cell configurations. Eventually, when the A/C ratio becomes <1 after the transition state, the anode is then the limiting electrode and dominates the irr-CE of AFLMB, which is comparable to the aforementioned A/C < 1 case that the irr-CE can be separated into cross-talk effects and dead Li+ sub. SEI. Thus, considering the effect of A/C ratio on irr-CE, when the A/C ratio is >1, the proportion of irr-CE from the anode is not observable and plotted in the light color bar with a dashed outline in Fig. 3g. When in the transition state, some of the irr-CE from dead Li formation will be compensated by excess Li as shown in the brown bar in Fig. 3g. At last, when the A/C ratio is <1, the irr-CE of AFLMB dominated by dead Li and subsequent SEI formation.

**Example 1: 1 M LiPF$_6$ in EC:DEC under 0.2 mA cm$^{-2}$.** Figure 4 shows the corresponding irreversible CE at the first, the second, and the tenth cycle of Li/Cu, NMC/Li, and NMC/Cu cells under the current density of 0.2 mA cm$^{-2}$ in the commercial electrolyte (1 M lithium hexafluorophosphate (LiPF$_6$) in EC:DEC). From the result obtained from the integrated protocol, the irr-CE of the first and the second cycle of Li/Cu cell is 8.36% and 5.79%, respectively, suggesting the 2.57% of higher irr-CE than second cycle is mainly caused by the first SEI formation, where the rest of that can be considered as dead Li+ sub. SEI formation based on the protocol discussed in Fig. 3g. For cathode/Li cell (Fig. 2a) with A/C = 50, the irr-CE of 10.36% in first cycle can be ascribed to the first intrinsic irr-capacity of cathode with the corresponding Ox. E.D.,

and with that of 0.9% in the second cycle to subsequent Ox. E.D. In the subsequent cycles, the fraction of cathode degradation is successfully extracted and quantified as 0.076%, with the rest of 0.36% attributed to subsequent oxidative electrolyte decomposition. In addition, it should be stated that the cycling performance of NMC/Li is only independent of Li when the amount of active Li is sufficient enough with low charge/discharge rate. When increasing the current density to 0.4 mA cm$^{-2}$, it is found that the capacity is declining quickly owing to the cathode degradation induced from soft short circuit and severe dendrite formation, as shown in Supplementary Fig. 9 and 10.

In the NMC/Cu cell, the first cycle irr-CE of NMC/Cu (AFLMB) cell is comparable to NMC/Li cell and can be described as the first irr-capacity of cathode, as the A/C ratio is currently >1 and the cathode is the limiting electrode, making the irr-CE from anode invisible. In the second cycle, the irr-CE is decreased to 4.48%, with the contribution mainly from dead Li+ sub. SEI formation as the cell is in the transition state to A/C < 1 and the irr-capacity and CE are limited by the anode. However, the higher irr-CE of AFLMB than Li/Cu cell in the tenth cycle may be owing to more complicated cell chemistry due to cross-talk effect[43–45]. Thus, the irr-CE can be separated into 5.67% of dead Li+ sub. SEI formation and 8.59% of cross-talk effects, respectively. Yet, it can still be concluded that dead Li formation is undoubtedly the most significant obstacle currently for the development of high-performance AFLMBs. In addition, experimental validation for the results obtained from the protocol is performed by measuring the amount of inactive Li in both Li/Cu and NMC/Cu cells at different cycles (Supplementary Discussion and Supplementary Fig. 5). It is found that the fraction of first extra SEI extracted from the protocol is very close to the practical SEI value measured from the TGC method. Moreover, the fraction of inactive Li+ in the NMC/Cu cell is found larger than that in Li/Cu cell from the first cycle, confirming the fraction of SEI in NMC/Cu cell is affected by cross-talk effects, thus, larger than that in Li/Cu cell. To conclude, the results of TGC measurements confirm the validity of our proposed protocol on dissecting the irr-CEs in both Li/Cu and cathode/Cu cells.

**Example 2: 1 M LiPF$_6$ in EC:DEC under 0.4 mA cm$^{-2}$.** The effect of current density is also studied using the integrated protocol. Figure 5 shows the obtained results using the same electrolyte from the previous section but under a current density of 0.4 mA cm$^{-2}$. The first, second, and tenth cycle irr-CE of Li/Cu cell is 7.5%, 6.2%, and 8.6%, respectively, representing 1.3% of first SEI formation on Cu with 6.2% of dead Li+ sub. SEI formation in the first cycle and mainly owing to dead Li+ sub. SEI formation in the subsequent cycles. Although the irr-CE of 11.19% for NMC/Li cell in the first cycle can be attributed to the first irr-capacity of NMC and that of 0.57% in the second cycle to subsequent oxidative electrolyte decomposition. As for the cathode degradation in the subsequent cycles, the fraction has been identified and calculated as 0.1%, which is larger than that in 0.2 mA cm$^{-2}$ case owing to the higher degradation rate under higher current density. Finally, after integrating the results from Li/Cu and NMC/Li cells, the irr-CE of NMC/Cu cell in the first cycle can be summarized as 11.68% first intrinsic irr-capacity of NMC as A/C ratio is larger than one and the determining electrode is the cathode. In the second cycle, the irr-CE is significantly larger than that in 0.2 mA cm$^{-2}$. This can be explained by the increased dendritic and dead Li formation rate and more serious SEI fracture under higher current density. In particular, the fraction of cross-talk effects is determined by subtracting the total irr-CE with that of Li/Cu cell in the second cycle as 4.04%. To conclude, the contribution of dead Li+ sub. SEI formation in irr-CE is

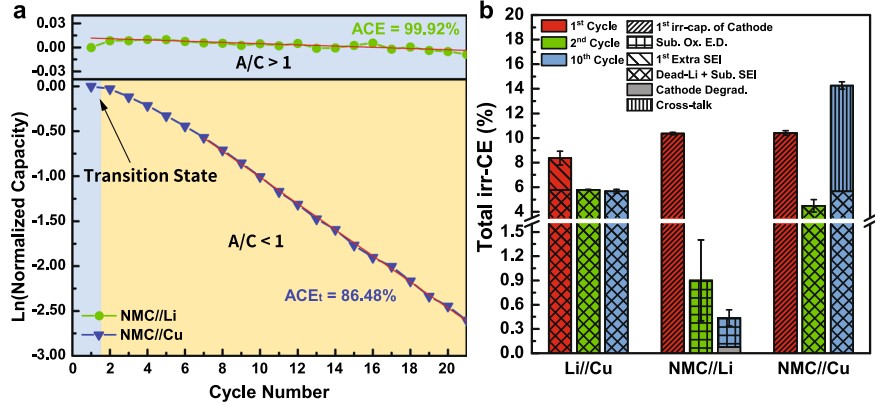

**Fig. 4 Results obtained from integrated protocol using 1 M LiPF$_6$ in EC:DEC (1:1) as electrolyte under the current density of 0.2 mA cm$^{-2}$. a** Normalized discharge capacity versus cycle number of NMC/Li and NMC/Cu cells. **b** irreversible CE comparison of Li/Cu, NMC/Li, and NMC/Cu cells. Error bars represent standard deviation, $n = 3$ independent replicates. The capacity retention comparison of NMC/Li and NMC/Cu cells are shown in Supplementary Fig. 8a.

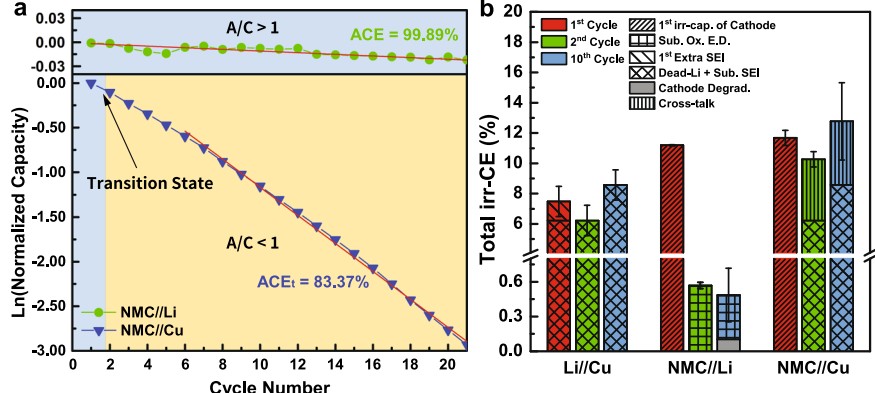

**Fig. 5 Results obtained from integrated protocol using 1 M LiPF$_6$ in EC:DEC (1:1) as electrolyte under the current density of 0.4 mA cm$^{-2}$. a** Normalized discharge capacity versus cycle number of NMC/Li and NMC/Cu cells. **b** irreversible CE comparison of Li/Cu, NMC/Li, and NMC/Cu cells. Error bars represent standard deviation, $n = 3$ independent replicates. The capacity retention comparison of NMC/Li and NMC/Cu cells are shown in Supplementary Fig. 8b. The charge/discharge profiles of each cell configuration are shown in Supplementary Fig. 12.

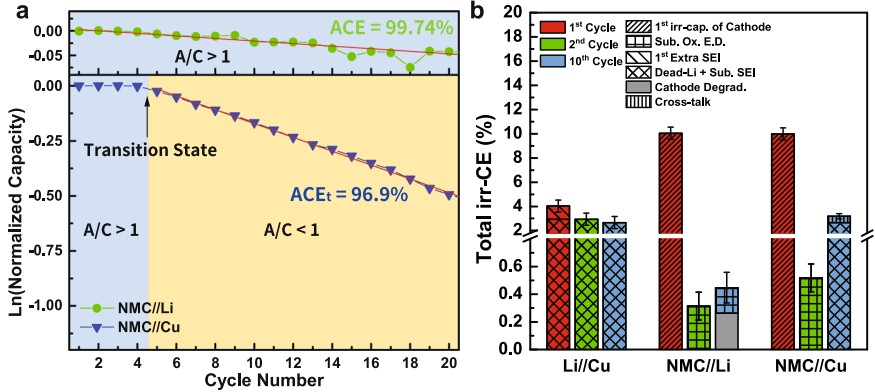

**Fig. 6 Results obtained from integrated protocol using 1 M LiPF$_6$ in EC:DEC (1:1) with 5% FEC added as electrolyte under the current density of 0.2 mA cm$^{-2}$. a** Normalized discharge capacity versus cycle number of NMC/Li and NMC/Cu cells. **b** irreversible CE comparison of Li/Cu, NMC/Li, and NMC/Cu cells. Error bars represent standard deviation, $n = 3$ independent replicates. The capacity retention comparison of NMC/Li and NMC/Cu cells are shown in Supplementary Fig. 8c. The charge/discharge profiles of each cell configuration are shown in Supplementary Fig. 13.

higher and consumes more amount of excess Li in each cycle, causing the faster transition to A/C < 1 and higher irr-CE of the second cycle than 0.2 mA cm$^{-2}$, and the major sources of irr-CE in AFLMB are still originated from the anode, namely dead Li+ sub. SEI formation and cross-talk effects.

**Example 3: 1 M LiPF$_6$ in EC:DEC with 5% FEC**. Figure 6 shows the results obtained from the integrated protocol using 1 M LiPF$_6$ in EC:DEC with 5% FEC as the electrolyte under the current density of 0.2 mA cm$^{-2}$. The first and second cycle irr-CE of Li/ Cu is 4.02% and 2.94%, respectively. Similar to the previous case,

the 1.08% difference can be attributed to the first SEI formation on Cu, and the rest of 2.94% can be attributed to dead Li+ sub. SEI formation. From the result of Li/Cu cell, the effect of FEC on suppressing dead Li and dendrite formation can already be revealed from the lower irr-CE than that of without FEC additive. Meanwhile, the irr-CE of NMC/Li cell in first cycle can be explained as the first intrinsic irr-capacity of NMC with the corresponding oxidative electrolyte decomposition (10.04%) and that in second cycle as subsequent oxidative electrolyte decomposition (0.31%). Meanwhile, the fraction of cathode degradation is determined as 0.26% from equation (1) in the supporting information, with the rest of 0.19% ascribing to electrolyte decomposition. Finally, the irr-CE of NMC/Cu in the first cycle, same as that of NMC/Li cell, can be attributed to 9.98% of the first intrinsic irr-capacity of NMC with oxidative electrolyte decomposition when the A/C ratio >1. However, the irr-CE of the second cycle is 0.52% and significantly lower than that in the NMC/Cu cell without FEC added. This can be again explained by the effect of FEC suppressing dead Li+ sub. SEI formation, leading to slower consumption of excess active Li on Cu. Thus, the A/C ratio of the cell is sustained greater than one longer than that without FEC additive before transiting to A/C < 1, and the irr-CE of the first four cycles is very low, as shown in Fig. 6a. In other words, the transition state is delayed to the fifth cycle. Meanwhile, the discharge capacity is found without decaying in the first four cycles, which also proves that the A/C > 1 region is extended to four cycles. As a result, the irr-CEs from the second to fourth cycle can be determined as subsequent oxidative electrolyte decomposition at the cathode like in the NMC/Li cell, different from those without FEC added. Later on, after the transition state and the A/C becomes <1, the increased irr-CE of 3.18% can be thus attributed to mainly dead Li+ sub. SEI formation (2.66%) and partially cross-talk effects (0.52%) at the anode.

To conclude, if we compare the results obtained from the proposed protocol between with/without FEC as an additive, the lower initial overpotential and polarization shown by Li/Li and Li/Cu cells after the introduction of FEC reveals the effect of FEC on forming better SEI and favoring Li nucleation on Cu. The lower irr-CE of Li/Cu cell and significantly lower irr-CE after the first cycle in NMC/Cu cell indicate the ability of FEC to suppress dendrite and dead Li. Interestingly, the irr-CE of NMC/Li cell does not show a significant difference with/without FEC, suggesting the contribution of FEC is less notable for the cathode. Although it has been reported that the addition of FEC into electrolytes can lead to a more compacted and stable LiF-rich SEI, which is also beneficial to the uniform Li deposit and better electrochemical performance[46]. By summarizing all the contributions from the above configuration within the proposed protocol, we can quantitively unravel the effects of FEC on mitigating the formation of dead Li and forming stable SEI in AFLMB.

## Discussion

In summary, we revealed the formation of dead Li and dendritic Li alongside the ISC via in situ OM and proposed the mechanism of Li deposition/dissolution on Cu. Furthermore, by combining the information and irr-CE of four different cell set-ups, an integrated protocol is proposed to unravel the concealed messages of various irr-CE. From the demystified information, the origin and proportion of irreversible coulombic efficiency in AFLMBs induced from first extra SEI, dead Li and subsequent SEI, cross-talk effects, first cycle intrinsic irreversible capacity of cathode, and subsequent oxidative electrolyte decomposition can be quantified and revealed. Although AFLMBs suffer from high

irr-CE and quick capacity fading and are often recognized as a poor cell configuration, AFLMB could also serve as an indispensable key in facilitating the development of better electrolytes and evaluating the performance of LMBs. In general, the integrated protocol proposed here can be further expanded to comprehensively examine the effectiveness of various strategies on improving the electrochemical performance of LMBs or AFLMBs, providing an insightful understanding of the behaviors of LMBs and AFLMBs and paving the way to the realization of next-generation high-energy rechargeable lithium batteries involving metal deposition/stripping chemistry.

## Methods

**Materials**. The commercial NMC electrodes (Amita Technologies Inc. Taiwan) contain 88.5 wt% of active material ($LiNi_{1/3}Co_{1/3}Mn_{1/3}O_2$, NMC-111), 4 wt% of polyvinylidene fluoride binder, and 7.5 wt% of conductive carbon (2.5 wt% of Super P with 5 wt% of KS6). The nominal areal discharge capacity of NMC electrodes used in this work is 2 mAh cm$^{-2}$.

Cu foil was used as an anode electrode in the anode-free cell. The Cu foil was cut into 19 mm in diameter, washed by 1 M hydrochloric acid for 10 min in an ultrasonic cleaner, followed by rinsing with deionized water and acetone three times, and finally vacuumed in the desiccator for 30 min before use.

**Cell assembly**. The OM Li/Cu cell was assembled by sealing a Li foil and copper foil at a distance ~250 μm in-between inside a plastic pouch cell. Copper wire was used as a terminal for both electrodes and the external circuit for electrochemical measurements. The OM cells and CR2032 coin cells were all assembled in an argon-filled glovebox (UNIlab Plus Glove Box, MBRAUN) where the oxygen and moisture content were kept <1 ppm.

Li foil (~300 μm, FMC Corporation) was attached on a 500 μm spacer and used as an anode in NMC/Li cell, identical electrodes in Li/Li symmetric cell, and tri-layer Celgard 2325 PP/PE/PP membrane as a separator. The NMC/Cu cells were assembled by pairing NMC electrode with a copper foil, with an 800 μm spacer to minimize the thickness difference of assemblies within the cell between NMC/Cu and NMC/Li cells to unify the cell pressure and make the results obtained from each cell configuration the most reliable. The electrolyte used was 1 M LiPF$_6$ in the mixture of ethylene carbonate (EC) and diethyl carbonate (v:v = 1:1) (Sigma-Aldrich) with or without 5% of fluoroethylene carbonate.

**Dead Li quantification**. Dead Li quantification was performed using TGC method reported by Meng et al.[32], Li/Cu cells at their stripped state were first disassembled inside an Ar-filled glovebox ($H_2O$, $O_2$ < 1 p.p.m.), then the Cu foil with residual inactive Li and separator were transferred into a 20 ml vial. The vial was sealed by a plastic lid with a rubber septum in the middle, then tightly wrapped with parafilm to prevent any gas leakage when $H_2$ gas was generated later. The pressure inside the vial was adjusted to 1 atm (0 mbar inside the glovebox environment) before the sealing process. After transferring the vial out of the glovebox, 0.5 ml of water was injected into the vial to allow a reaction with dead Li. The added excess amount of water would fully react with inactive metallic Li, generating lithium hydroxide and $H_2$ gas. After complete reaction and $H_2$ gas formation, a gas-tight syringe was used to take 250 μl of gas within the vial and to inject it into the GC for $H_2$ measurement. The calibration line of GC was measured by weighing several different weights of metallic Li and measured the respective $H_2$ areas as a function.

**In situ OM observation**. In situ plating and stripping of Li on copper foil was observed using OM in a plastic pouch cell by applying 500 mA cm$^{-2}$ of charge/discharge current density, as shown in Fig. 1a and the movies are also included in the supporting information. The electrochemical measurement of the cell was simultaneously conducted on a PGSTAT101 Autolab potentiostat (Metrohm), with the potential and current resolution of 3 μV and 10 nA, respectively.

**Electrochemical tests**. Galvanostatic charge/discharge was used to cycle batteries with an applied current density of either 0.2 or 0.4 mA cm$^{-2}$ to study the effect of charge/discharge rate. Galvanostatic charge/discharge tests were performed on Arbin BT-2000 (Arbin Instruments) battery test equipment at room temperature.

## Data availability

The data that support the findings of this study are available from the authors on reasonable request. Correspondence and requests for materials should be addressed to B.-J.H.

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

## Acknowledgements

Financial support from the Ministry of Science and Technology of Taiwan (MOST 109-2639-E-011-001-ASP, 109-2923-E-011-008, 109-3116-F-011-CC1, 109-2124-M-002-008, 109-2923-E-011-009, 109-2221-E-011-063-MY3), the Ministry of Education of Taiwan (U2RSC program, MOE 1080059, Taiwan's Deep Decarbonization Pathways toward a Sustainable Society Project (AS-KPQ-106- DDPP) from Academia Sinica as well as the supporting facilities from National Taiwan University of Science and Technology (NTUST) and National Synchrotron Radiation Research Centre (NSRRC) are all gratefully acknowledged.

## Author contributions

C.-J.H. and B.J.H. conceived and designed the work; C.-J.H., B.T., H.-C.T., K.N.S., H.S., T.T.H., T.T.B., and L.-M.K. performed the experiments; C.-J.H., B.T., H.-C.T., H.S., T.T.H., T.T.B., L.-M.K., W.-N.S., S.-H.W., and B.J.H. analyzed the data; C.-C.W. helped measure and process the TXM data; C.-J.H. and B.J.H. wrote the paper. All the authors discussed the results and commented on the manuscript. All authors have approved the final version of the manuscript.

## Competing interests

The authors declare no competing interests.
