## [Peer Review File · Nature Communications]

Reviewer #1 (Remarks to the Author):

This paper proposes a method to analyze the irreversible capacity loss mechanism of anode-free lithium metal batteries. First, the authors observe the lithium deposition and stripping modes in a Li//Cu cell using in situ optical microscope analysis. The inhomogeneous lithium deposition/stripping and the significant formation of “dead Li” are clearly observed. In the next part, a method to analyze the irreversible capacity loss of the NMC//Cu cell is explained. The irr-capacity loss at the NMC cathode and Cu anode current collector are assessed by the value obtained from the NMC//Li and Li//Cu cells. In the last part, the proposed method is applied to the analysis of NMC//Cu cells that have different applied current density and electrolyte additive.

The anode-free lithium battery is one of the most promising cell configurations that can provide an extremely high energy density. However, irreversible lithium deposition/stripping that results in low coulombic efficiency and poor cycle performance has prevented it to be a major option yet. Therefore, the subject of this paper is interesting and timely. However, the proposed method is based on a series of presumptions and over-simplified mechanism analysis.

For example, in the analysis of Li//Cu cells on page 4, the authors state that “...it is mainly contributed to dead Li formation (green bar in Fig. 3g) after the formation of a stable SEI in the 1st cycle, especially at low C-rate. Therefore, the irr-CE originated from the reductive electrolyte decomposition (red bar in Fig. 3g) can be calculated by subtracting the value in the 1st cycle with that of the 2nd cycle from the Li//Cu cell protocol.” In conventional electrolytes, the SEI formed on the surface of Cu or Li metal is not quite stable. Therefore, it is more reasonable to presume that the electrolyte decomposition and SEI formation still account for a significant fraction of the 2nd cycle irreversible capacity loss. In addition, there is no guarantee that the amount of the dead lithium from the 2nd cycle would be the same as that from the 1st cycle; the value from the 2nd cycle cannot simply substitute for the value for the 1st cycle.

Another example can be found on page 5 in the discussion of NMC//Li cells. The origins of the irreversible capacity loss of NMC//Li cells are attributed to (1) intrinsic irr-capacity, (2) CEI (cathode electrolyte interface) layer formation from, and (3) oxidative electrolyte decomposition at high voltage. And it is assumed that the mechanisms (1) and (2) are negligible from the 2nd cycle. Based on these premises, the authors suggest that the initial irr-capacity loss only by the mechanisms (1) and (2) in the 1st cycle can be calculated by subtracting the irr-capacity value in the 2nd cycle from that in the 1st cycle. However, the CEI formation and the oxidative electrolyte decomposition are correlated with each other as the CEI layer is the product of the decomposition reaction. And it is not realistic to assume that the intrinsic irr-capacity loss of cathodes only exists during the first cycle.

Therefore, the proposed analysis of NMC//Cu cells that are based on the analysis of the NMC//Li and Li//Cu cells are not convincing enough. Without an experimental method that enables to evaluate the effect of each irr-capacity loss mechanism quantitatively, it is difficult to prove the validity of the proposed method. This manuscript doesn't meet the high standards of the Nature Communications in terms of scientific rigorousness and practical impact.

Reviewer #2 (Remarks to the Author):

The authors systematically studied four different types of cell configuration, including Li//Li symmetric cells, Li//Cu cells, cathode/Li cells, and cathode/Cu anode-free cells, to unfold the intrinsic reasons and contributions of individual irreversible coulombic efficiency (irr-CE) in LMBs or AFLMBs by titration gas chromatography (TGC) method. This work might attract broad interests in the related field. I recommend the manuscript to be accepted for publishing but with a major revision. Here are the points need to be carefully addressed.

- 1) According to Y. S. Meng, Nature volume 572, pages 511–515 (2019), their work demonstrated the formation mechanism of inactive Li in different electrolytes and underlies the cause of low CE of Li anodes by TGC method. The authors need to work and think hard to give a clear differentiate between Meng's work and your work, and point out your novelty and significance.
- 2) Since this work adopted the TGC method to quantify dead Li and this method maybe not familiar with the majority readers, the authors need to give some introduction on the mechanism to help readers know this technology better. And how did the authors achieve the proportions for dead Li, red E. D. and Ox E. D. in Figure 3-7? I cannot find clear descriptions on it and this seems important to support the conclusions of this work. So, the authors should clarify this point to the readers clearly.
- 3) When we look into the irr CE data of Li/Cu and NMC/Cu cells, why the dead Li problem are more serious in NMC/Cu cells than that in Li/Cu cells, leading to big loss of CE performance? The authors should give further explanation on it.
- 4) For the data of NMC/Li cells, I notice that the dead Li problem seems be alleviated, why. And how did the authors collect dead Li from Li foil and ensure all the dead Li are removed completely from it? The authors should give further details on it because this matters the results.
- 5) This work seems to emphasize on the fundamental understanding of Li behaviors, so the studied system cannot be limited in ester electrolyte system, but it should be extended to ether system to offer a general understanding in the both popular electrolytes. Based on this, it would be better to give some data in ether systems and extend the achieved conclusions and principles in ether-friendly battery systems.
- 6) Partial descriptions should be concisely phrased and more scientific, e.g. "just like a rose always has its thorns"
- 7) Some related contributions can be referred to help the better express of this article.
 - i) For the current strategies for Li metal anodes in Introduction part, recommend citing "Tuning wettability of molten lithium via a chemical strategy for lithium metal anodes, Nature Communications, 2019, 10, 1-8" and "A 3D Lithium/Carbon Fiber Anode with Sustained Electrolyte Contact for Solid-State Batteries, Advanced Energy Materials, 2019, 10 (3), 1903325."
 - ii) For the Li deposition/dissolution part, recommend citing "Towards Better Li Metal Anodes: Challenges and Strategies, Materials Today, 2020, 33, 56-74."

Firstly, the authors express their appreciation to the editor and the reviewers for careful and in-depth reading of this manuscript and for the thoughtful comments and constructive suggestions, which help us to improve the quality of this manuscript. Each comment has been carefully considered and responded on a point-by-point basis.

Point-to-Point Response (response in blue font)

Reviewer #1:

This paper proposes a method to analyze the irreversible capacity loss mechanism of anode-free lithium metal batteries. First, the authors observe the lithium deposition and stripping modes in a Li//Cu cell using in situ optical microscope analysis. The inhomogeneous lithium deposition/stripping and the significant formation of “dead Li” are clearly observed. In the next part, a method to analyze the irreversible capacity loss of the NMC//Cu cell is explained. The irr-capacity loss at the NMC cathode and Cu anode current collector are assessed by the value obtained from the NMC//Li and Li//Cu cells. In the last part, the proposed method is applied to the analysis of NMC//Cu cells that have different applied current density and electrolyte additive.

The anode-free lithium battery is one of the most promising cell configurations that can provide an extremely high energy density. However, irreversible lithium deposition/stripping that results in low coulombic efficiency and poor cycle performance has prevented it to be a major option yet. Therefore, the subject of this paper is interesting and timely. However, the proposed method is based on a series of presumptions and over-simplified mechanism analysis.

Reply:

Thank you for your recognition and comments. After carefully reading these comments, we have put great effort into performing more experiments and modifying our manuscripts based on our own experimental results to address the reviewer’s concern. We think our modified protocol can serve as a rigorous platform for the performance evaluation and determination and quantification of the sources of irreversible coulombic efficiency in anode-free Li metal batteries. Therefore, we think our work will attract great

interests from the broad audience of Nature Communication. Please see the detailed point-to-point response as follows.

Comment #1: “For example, in the analysis of Li//Cu cells on page 4, the authors state that “...it is mainly contributed to dead Li formation (green bar in Fig. 3g) after the formation of a stable SEI in the 1st cycle, especially at low C-rate. Therefore, the irr-CE originated from the reductive electrolyte decomposition (red bar in Fig. 3g) can be calculated by subtracting the value in the 1st cycle with that of the 2nd cycle from the Li//Cu cell protocol.” In conventional electrolytes, the SEI formed on the surface of Cu or Li metal is not quite stable. Therefore, it is more reasonable to presume that the electrolyte decomposition and SEI formation still account for a significant fraction of the 2nd cycle irreversible capacity loss. In addition, there is no guarantee that the amount of the dead lithium from the 2nd cycle would be the same as that from the 1st cycle; the value from the 2nd cycle cannot simply substitute for the value for the 1st cycle.”

Reply:

Thank you for your comment. In order to answer the reviewer’s concerns, we have performed TGC method¹ to quantify the amount of dead Li and SEI after one, two, three, four, and five cycles of Li//Cu cells, respectively. It is found that although the small fraction of SEI slightly increased in each cycle, the irr-CEs in the subsequent cycles are quite similar, as shown by the average Li loss in each cycle after the 1st cycle in Fig. R1. From the obtained results, small fraction of SEI loss is still present in the 2nd cycle and subsequent cycles, as the reviewer’s concern, which can be explained by the instability of SEI and the subsequent fracture due to dendrite formation². To conclude, it is not possible to easily decouple the irr-CEs from the contribution of dead Li and SEI formation in a Li//Cu cell without an additional experiment like a TGC measurement.

Therefore, based on the aforementioned results, we have modified our proposed protocol accordingly. Firstly, the irr-CE in Li//Cu cell in the subsequent cycles is considered as the combination of dead Li and subsequent SEI formation, denoted as dead Li + sub. SEI. Secondly, for the difference between the 1st cycle irr-CE and that of the 2nd cycle, based on the fact that the irr-CEs after the 1st cycle are quite similar to each other, suggesting an additional irreversible reaction in the 1st cycle, we attributed it

to the 1st cycle extra SEI formation (1st extra SEI) due to the initial reductive electrolyte decomposition on the Cu surface when the cell is first discharged.

Action:

(In the revised supporting information page 1-2)

Fraction of dead Li⁰ and SEI Li⁺ in Li//Cu cell determined by the TGC method

Due to the fact that it is unable to separate the fraction of dead Li and SEI simply from the electrochemical results of Li//Cu cell. A precise quantification method like TGC measurement¹ is required for determining the fractions of dead Li and SEI in each cycle of the Li//Cu cells. Thus, we have measured the proportion of dead Li and SEI in each cycle from one to five cycles of Li//Cu cells by the TGC method (See **Fig. S2**). The results of the TGC measurements and electrochemical tests show that after the 1st cycle, the irr-CEs of Li//Cu cell are quite comparable in the subsequent cycles, with the fraction of subsequent SEI gradually increased in each cycle. This can be explained as the instability of SEI and the subsequent fracture due to dendrite formation². To conclude, since the proportion of dead Li and subsequent SEI needs to be quantified by TGC measurements, the two origins of irr-CE are combined in the proposed protocol and denoted as dead Li + sub. SEI. As for the difference between the 1st cycle irr-CE and that of the 2nd cycle, since that the irr-CEs after the 1st cycle are quite similar to each other, indicating an additional irreversible reaction in the 1st cycle, we attributed it to the 1st cycle extra SEI formation due to the initial reductive electrolyte decomposition on the Cu surface when the cell is first discharged, namely 1st extra SEI in the protocol.

Fig. R1. TGC measurements quantifying the amount of dead Li⁰ and SEI Li⁺ associated with the total capacity loss of Li//Cu cells after one, two, three, four, and five cycles (the 1st, 2nd, 3rd, 4th, 5th). (Fig. S2 in Supporting Information)

Comment #2: “Another example can be found on page 5 in the discussion of NMC//Li cells. The origins of the irreversible capacity loss of NMC//Li cells are attributed to (1) intrinsic irr-capacity, (2) CEI (cathode electrolyte interface) layer formation from, and (3) oxidative electrolyte decomposition at high voltage. And it is assumed that the mechanisms (1) and (2) are negligible from the 2nd cycle. Based on these premises, the authors suggest that the initial irr-capacity loss only by the mechanisms (1) and (2) in the 1st cycle can be calculated by subtracting the irr-capacity value in the 2nd cycle from that in the 1st cycle. However, the CEI formation and the oxidative electrolyte decomposition are correlated with each other as the CEI layer is the product of the decomposition reaction. And it is not realistic to assume that the intrinsic irr-capacity loss of cathodes only exists during the first cycle.”

Reply:

Thank you for the constructive comments. Sorry for the confusion. We agree with the reviewer’s comment that the CEI formation and the oxidative electrolyte decomposition are correlated with each other as the CEI layer is the product of the decomposition reaction. Actually, we did not intend to distinguish these two in the original manuscript

either. To make it clear, we have modified our protocol to propose that both the aforementioned irreversible reactions account for a fraction in the subsequent irr-CEs, namely “sub. Ox. E.D.” in the proposed protocol. Furthermore, we have additionally dissected the fraction of cathode degradation from the irr-CE of NMC//Li cells through calculating average CE from the linear fitted slope of the normalized discharge capacity retention in the subsequent cycles.

For the intrinsic irreversible capacity, we focused on the 1st cycle irreversible capacity of layered oxide cathode materials as reported in the literature, which could be attributed to the slow lithium kinetics at high lithium contents and partially the formation of Li₂MO₂-like (M= transition metals) phases^{3, 4, 5}. This intrinsic capacity loss of layered oxide cathode materials reported only exists in the 1st cycle and could not be recovered.

We also confirmed the phenomenon by the first charging/discharging the NMC//Li cell at the normal cut-off voltage window (2.5 V to 4.3 V), in which the 1st intrinsic irreversible capacity is observed, then performing deep discharge in the 2nd cycle to recover the irreversible capacity (See **Fig. R2**) in Fig. S3 of the revised supporting information. When the normal cut-off voltage window was resumed in the 3rd cycle, the reversible capacity notably dropped back to that level in the 1st cycle. Therefore, it can be concluded that the intrinsic capacity loss of layered oxide NMC only exists in the 1st cycle since even if deep discharge was performed in the 2nd cycle, the reversible capacity of both 2nd and 3rd cycles within the normal cut-off voltage window is still the same with that of the 1st cycle. The capacity loss in the following cycles should be attributed to the combination of cathode degradation and subsequent oxidative electrolyte decomposition (sub. Ox. E.D.), when the reversible capacity starts declining. In particular, the fraction of cathode degradation after the reversible capacity starts decaying can be calculated from the slope of the fitted line of the normalized discharged capacity retention based on equation (1) in supporting information. Since the reversible capacity loss of cathode//Li cell is directly related to the cathode degradation, thereby the fading rate can be correlated to the irr-CE of cathode degradation in each cycle. Thus, after quantifying the irr-CE of cathode degradation, the fraction of sub. Ox. E.D. within the capacity fading region could be singled out and quantified as the difference between the total irr-CE and that of cathode degradation.

Action:

Fig. R2 (Fig. S3 in the revised supporting information) has been included in the revised supporting information page 2.

Fig. R2. Charge and discharge curves of NMC//Li cell with deep discharge in the 2nd cycle and normal discharge in the 1st and 3rd cycles, respectively. (Fig. S3 in the revised supporting information)

Comment #3: Therefore, the proposed analysis of NMC//Cu cells that are based on the analysis of the NMC//Li and Li//Cu cells are not convincing enough. Without an experimental method that enables to evaluate the effect of each irr-capacity loss mechanism quantitatively, it is difficult to prove the validity of the proposed method.

Reply:

Thank you for the comment. We have performed additional experiments to validate our proposed protocol for the analysis of NMC//Cu cells, as discussed above, including the validation of irr-CEs in Li//Cu cells and NMC//Li cells. After modifying our proposed protocol based on the reviewers' comments and suggestions, we think it is now rigorous enough to serve as a platform for the evaluation of AFLMBs and can be widely applied to various systems for the development of next-generation high energy and safe (AF)LMBs.

Reviewer #2 :

The authors systematically studied four different types of cell configuration, including Li//Li symmetric cells, Li//Cu cells, cathode/Li cells, and cathode/Cu anode-free cells, to unfold the intrinsic reasons and contributions of individual irreversible coulombic efficiency (irr-CE) in LMBs or AFLMBs by titration gas chromatography (TGC) method. This work might attract broad interests in the related field. I recommend the manuscript to be accepted for publishing but with a major revision. Here are the points need to be carefully addressed.

Reply:

Thank you very much for your comment. Based on the reviewer's comments, we have performed more experiments and modified our manuscript to address the novelty and significance of our work to the broad audience of Nature Communication. Please see the detailed point-to-point response as follows.

Comment #1: According to Y. S. Meng, Nature volume 572, pages511–515(2019), their work demonstrated the formation mechanism of inactive Li in different electrolytes and underlies the cause of low CE of Li anodes by TGC method. The authors need to work and think hard to give a clear differentiate between Meng's work and your work, and point out your novelty and significance.

Reply:

Thank you for the constructive suggestion. We have rigorously discussed the difference between our and Meng's work and highlighted our contribution based on the proposed protocol in the revised manuscript.

In Meng's work, the dominant source of capacity loss and inactive Li was successfully identified as dead-Li by TGC method using the Li//Cu cell configuration. They comprehensively discussed the formation of inactive Li and determine the origin of irr-CE of Li anodes within Li//Cu cells by decoupling the dead Li and SEI formation, which provides new strategies for more efficient Li plating and stripping on Cu substrate. However, the important information of the irr-CE or capacity loss from the cathode, and cross-talk effects in cathode//Li or anode-free cathode//Cu cells could not be demystified in the Li//Cu cells from the TGC method. Thus, we proposed an integrated

protocol which could systematically identify the origins of irr-CE in various cell configurations, e.g., Li//Cu, cathode//Li, and cathode//Cu cells, from not only Li anodes in Li//Cu cells like the TGC method, but also the cathodes in cathode//Li half-cells. Furthermore, the cause of low irr-CE of AFLMBs can also be determined at different states of the batteries through the proposed protocol. To conclude, our work provides an overall understanding and quantification to the irr-CEs from the full spectrum of different cell configurations such as 1st extra SEI, dead Li and subsequent SEI, cross-talk effects, 1st cycle intrinsic irreversible capacity of cathode, and subsequent oxidative electrolyte decomposition. Although the precise determination of dead Li and SEI still requires other experimental approaches like the TGC method, our proposed protocol provides an overall perspective on evaluating the performance of LMBs and AFLMBs by identifying and quantifying the sources of various irr-CEs.

Action:

(In the revised manuscript page 2)

“One efficient way is to study the irreversible coulombic efficiency (irr-CE), which may represent the side reactions and sources of capacity loss in the battery. Meng et al. demonstrated an analytical method of titration gas chromatography (TGC) to quantify the contribution of dead Li to the total irr-CE in Li//Cu cells, identifying dead Li as the major reason accounted for the capacity loss of Li//Cu cells¹. They comprehensively discussed the formation of inactive Li and determine the origin of irr-CE of Li anodes within Li//Cu cells by decoupling the dead Li and SEI formation, which provides new strategies for more efficient Li plating and stripping on Cu substrate. However, the important information of the irr-CE or capacity loss from the cathode, and cross-talk effects in cathode//Li or anode-free cathode//Cu cells could not be extracted from the TGC method. Thus, there is still a lack of holistic methodology to identify and quantify the irr-CEs in LMBs and AFLMBs.” (In the revised manuscript page 2)

“In this work, we systematically study four different types of cell configuration, including Li//Li symmetric cells⁶, Li//Cu cells, cathode//Li cells, and cathode//Cu anode-free cells, as an integrated protocol to unfold the intrinsic reasons and contributions of individual irr-CEs from not only Li anodes in Li//Cu cells, but also the cathodes in cathode//Li half-cells. Furthermore, the cause of low irr-CE of AFLMBs can also be determined at

different states of the batteries through the proposed protocol. Meanwhile, we also observed dendritic Li induced internal short-circuit and visualize the formation of dead Li in a Li//Cu cell using *in-situ* optical microscopy (OM) and transmission X-ray microscopy (TXM), and proposed the mechanism of Li nucleation and deposition/dissolution on Cu. This work provides an overall understanding and quantification to the irr-CEs from the full spectrum of different cell configurations such as 1st extra SEI, dead Li and subsequent SEI, cross-talk effects, 1st cycle intrinsic irreversible capacity of cathode, and subsequent oxidative electrolyte decomposition. The proposed protocol could serve as a platform form an overall perspective to evaluate the performance of LMBs and AFLMBs, and can be widely applied to various systems for the development of next-generation high-energy batteries.” (In the revised manuscript page 2)

Comment #2: Since this work adopted the TGC method to quantify dead Li and this method maybe not familiar with the majority readers, the authors need to give some introduction on the mechanism to help readers know this technology better. And how did the authors achieve the proportions for dead Li, red E. D. and Ox E. D. in Figure 3-7? I cannot find clear descriptions on it and this seems important to support the conclusions of this work. So, the authors should clarify this point to the readers clearly.

Reply:

Thank you for the constructive suggestion. We have added a section entitled of “Dead-Li quantification” in the “Methods” part of the manuscript to give a detailed introduction of the TGC method to the readers.

To answer the reviewer’s concerns on how to dissect all the irr-CEs in the protocol, we have provided a detailed flowchart which describes the procedures to obtain the proportion of different irr-CEs from each cell configuration and how to utilize them in anode-free cells with slightly modified names of each origin of irr-CE, namely 1st extra SEI formation, dead Li + sub. SEI, the 1st irr-capacity of the cathode (with Ox. E.D.), sub. Ox. E.D., and cathode degradation for better interpretation (see **Fig. R3 and R4**). It should be noted that we have modified the protocol by combining the dead Li formation and red E.D. in the subsequent cycles, namely dead Li + sub. SEI, due to the fact that it

is not possible to easily decouple the irr-CEs from the contribution of dead Li and SEI formation in a Li//Cu cell without an additional experiment like a TGC measurement.

In order to analyze the irr-CE in AFLMB, we need to first obtain the irr-CEs in Li//Cu and cathode//Li cells considering the reactions of cathode and anode in AFLMB.

- In a Li//Cu cell, we dissect the irr-CEs by subtracting the 1st cycle irr-CE from the 2nd cycle irr-CE to quantify the contribution of 1st cycle extra SEI formation due to the initial reductive electrolyte decomposition on the Cu surface when the cell is first discharged, namely 1st extra SEI in the protocol. Then, the irr-CE remained in the following cycles can be attributed to dead Li and the subsequent SEI formation due to the reductive electrolyte decomposition (Dead Li + Sub. SEI).
- In a cathode//Li cell, the 1st cycle irr-CE of cathode//Li cell can be explained as the 1st irr-capacity of the cathode with the additional oxidative electrolyte decomposition (Ox. E.D.) and the corresponding CEI formation. As for the subsequent cycles, the origins of irr-CEs can be separated into two sources. When the reversible capacity remains the same and stable, the irr-CE of the cell can be attributed to the subsequent oxidative electrolyte decomposition (Sub. Ox. E.D.) with the consequent CEI formation included; however, when the reversible capacity starts to fade, then the irr-CE would become the sum of cathode degradation (cathode degrad.) and sub. Ox. E.D.. To be more specific, the fraction of cathode degradation can be calculated from the slope of the fitted line of the normalized discharged capacity retention based on equation (1) ($\ln\left(\frac{Capacity_{(n-1)}}{Capacity_{(1)}}\right) = (n-1)\ln(Average\ CE)$); thus, the fraction of sub. Ox. E.D. within the capacity fading region could be quantified as the difference between the total irr-CE and that of cathode degradation.
- In a NMC//Cu cell (AFLMB), after identifying and quantifying the irr-CEs in Li//Cu and cathode//Li cells, the obtained irr-CEs can be further transfer into the NMC//Cu cell (AFLMB) at its different A/C ratio states. Due to the fact that A/C ratio would affect the determination of the irr-CE origins, identifying the A/C ratio of the cathode//Cu cell is, therefore crucial to the proposed protocol and could guide us through the steps in the flowchart to dissect the irr-CEs in cathode//Cu cell comprehensively (see **Fig. R3**). Hence, we propose to consider and quantify the loss of CE and capacity from a new aspect, i.e., the anode/cathode (A/C) ratio, in the AFLMBs.

When the initial A/C ratio is less than one in the 1st cycle of cathode//Cu cell, the 1st cycle irr-CE is controlled by anode electrode and can be separated into three components, namely 1st extra SEI formation, dead Li + sub. SEI, and cross-talk effects. The fraction of 1st extra SEI formation and dead Li + sub. SEI can be transferred from the 1st cycle irr-CE in Li//Cu cell; thus, that of cross-talk effects can be calculated by subtracting the sum of the 1st cycle irr-CE in Li//Cu cell from the total 1st cycle irr-CE in cathode//Cu cell. In the subsequent cycles, the irr-CE of cathode//Cu cell can be dissected into dead Li + sub. SEI (transferred from Li//Cu cell) and the cross-talk effects (calculated from the irr-CE difference between Li//Cu and NMC//Cu cell in the subsequent cycles).

Meanwhile, if the initial A/C ratio is larger than one, namely cathode becomes the limiting electrode, then the 1st cycle irr-CE is dominated by cathode electrode and equals to the fraction of the 1st irr-capacity of cathode. In the subsequent cycles, the irr-CE can be separated into cathode degradation and sub. Ox. E.D. when the A/C ratio remains larger than one, namely before the transition state. It should be noted that the fraction of cathode degradation in cathode//Cu cell could also be calculated from the slope in equation (1) $\left(\ln\left(\frac{Capacity_{(n-1)}}{Capacity_{(1)}}\right) = (n-1)\ln(Average\ CE)\right)$ like in cathode//Li cell; however, the value may not necessarily equal to that in cathode//Li cell considering the cathode degradation mechanism may be different among two cell configurations. However, as the active Li suffers from the continuous consumption by dead Li and subsequent SEI formation, the A/C ratio would eventually become less than one. Therefore, the anode is then the limiting electrode and dominates the irr-CE of cathode//Cu cell, which is comparable to the aforementioned $A/C < 1$ case that the irr-CE can be separated into cross-talk effects and dead Li + sub. SEI.

Action:

Dead-Li Quantification (in the revised manuscript page12)

Dead-Li quantification was performed using titration gas chromatography (TGC) method reported by Meng et. al., Li//Cu cells at their stripped state were first disassembled inside an Ar-filled glovebox ($H_2O, O_2 < 1$ p.p.m.), then the Cu foil with residual inactive Li and separator were transferred into a 20 ml vial. The vial was sealed

by a plastic lid with rubber septum in the middle then tightly wrapped with parafilm to prevent any gas leakage when H₂ gas was generated later. The pressure inside the vial was adjusted to 1 atm (0 mbar inside the glovebox environment) prior to the sealing process. After transferring the vial out of the glovebox, 0.5 ml of water was injected into the vial to allow reaction with dead-Li. The added excess amount of water would fully react with inactive metallic Li, generating lithium hydroxide and H₂ gas. After complete reaction and H₂ gas formation, a gas-tight syringe was used to take 250 μl of gas within the vial and to inject it into the GC for H₂ measurement. The calibration line of GC was measured by weighing several different weights of metallic Li and measured the respective H₂ areas as a function.

Procedures to obtain irr-CEs in different cell configurations (in the revised supporting information page 3)

The above mentioned discussion and explanation for the protocol flowchart has been written in for a section and included in the supporting information page 3

“In this section, the detailed step-by-step flowchart of the proposed protocol is provided as shown in **Fig. S3**.....

.....Therefore, the anode electrode is then the limiting electrode and dominates the irr-CE of cathode//Cu cell, which is comparable to the aforementioned $A/C < 1$ case that the irr-CE can be separated into cross-talk effects and dead Li + sub. SEI.”

Fig. R3. A step-by-step flowchart of the proposed protocol for identifying the origin of each irr-CE in different cell configurations. (Fig. S3 in the revised supporting information)

Modification of figures

Based on the reviewer's comments and the modified protocol, we have slightly modified our scheme of the proposed protocol and the figures of obtained results in different systems. For simplicity, we only show the modified figures in the following content, namely **Fig. R4-R7**.

Fig. R4. Integrated protocol and scheme of different cell configuration at fully charged and discharged states. **a, b, c,** Scheme of Li//Cu, NMC//Li, and NMC//Cu cells at fully charged/ Li plated state in the first cycle, respectively. **d, e, f,** Scheme of Li//Cu, NMC//Li, and NMC//Cu cells at fully discharge/ Li stripped state in the first cycle, respectively. **g,** Proposed integrated protocol to unravel the origins of irreversible CE in AFLMB by Li//Cu and cathode/Li cells. The blue shell on Li represents the SEI layer. Sub. SEI stands for the subsequent SEI formation, Ox. E.D. for oxidative electrolyte

decomposition, and Cathode Degrad. for cathode degradation, respectively. (Fig. 3 in the revised manuscript)

Fig. R5. Results obtained from integrated protocol using 1M LiPF_6 in EC:DEC (1:1) as electrolyte under the current density of 0.2 mA cm^{-2} . **a**, normalized discharge capacity versus cycle number of NMC//Li and NMC//Cu cells. **b**, irreversible CE comparison of Li//Cu, NMC//Li, and NMC//Cu cells. The capacity retention comparison of NMC//Li and NMC//Cu cells are shown in Fig. S8a. (Fig. 4 in the revised manuscript)

Fig. R6. Results obtained from integrated protocol using 1M LiPF_6 in EC:DEC (1:1) as electrolyte under the current density of 0.4 mA cm^{-2} . **a**, normalized discharge capacity

versus cycle number of NMC//Li and NMC//Cu cells. **b**, irreversible CE comparison of Li//Cu, NMC//Li, and NMC//Cu cells. The capacity retention comparison of NMC//Li and NMC//Cu cells are shown in Fig. S8b. The charge/discharge profiles of each cell configuration are shown in Fig. S12. (**Fig. 5** in the revised manuscript)

Fig. R7. Results obtained from integrated protocol using 1M LiPF₆ in EC:DEC (1:1) with 5% FEC added as electrolyte under the current density of 0.2 mA cm⁻². **a**, Normalized discharge capacity versus cycle number of NMC//Li and NMC//Cu cells. **b**, irreversible CE comparison of Li//Cu, NMC//Li, and NMC//Cu cells. The capacity retention comparison of NMC//Li and NMC//Cu cells are shown in Fig. S8c. The charge/discharge profiles of each cell configuration are shown in Fig. S13. (**Fig. 6** in the revised manuscript)

Comment #3: When we look into the irr CE data of Li//Cu and NMC/Cu cells, why the dead Li problem are more serious in NMC/Cu cells than that in Li//Cu cells, leading to big loss of CE performance? The authors should give further explanation on it.

Reply:

Thank you for the comment. Due to the fact that the only difference between Li//Cu and NMC//Cu cells is the active Li source within the system, the major cause for the higher irr-CE of NMC//Cu cells than that of Li//Cu cells should be attributed to the cross-talk effects from cathode. Therefore, the Li plating/stripping chemistry would be greatly

affected by the crossover of transition metal ions from the cathode materials upon cycling. Thus, the SEI formation mechanism and composition could be substantially altered and the morphology of Li deposits would also be greatly affected^{7, 8}, leading to more complicated reactions and higher irr-CE at the anode electrode.

Action:

“Nevertheless, it should be noted that apart from the sources of irr-CEs and capacity loss in Li//Cu and cathode//Li cells would affect the evaluation of AFLMB, cross-talk effects would also account for the addition irr-CE upon the cycling of AFLMB. During the cycling of AFLMB, cross-talk effects could take place, namely the crossover of transition metal ions from the cathode materials to anode, leading to greatly altered Li plating/stripping chemistry as well as the SEI formation mechanism^{7, 8}. In other words, more complicated side reactions and higher irr-CE of the Li plating/stripping processes than those in Li//Cu cell may occur. Therefore, when dissecting the irr-CEs of AFLMB, cross-talk effects should be also considered.” (in the revised manuscript page 6.)

Comment #4: For the data of NMC/Li cells, I notice that the dead Li problem seems be alleviated, why. And how did the authors collect dead Li from Li foil and ensure all the dead Li are removed completely from it? The authors should give further details on it because this matters the results.

Reply:

Thank you for the comments. We think maybe there is some misunderstanding on the results of NMC/Li cells. Since Li foil is used in NMC/Li half-cells, with the excess amount of Li from the Li foil, the dead-Li problem is thus invisible in this system and would not affect the irr-CE in NMC//Li cells. The irr-CEs in NMC//Li cells mainly originate from oxidative electrolyte decomposition (with the correlated CEI formation) and cathode degradation (See **Fig. R5**). As a result, we did not consider the fraction of dead Li and measure the amount of dead Li within NMC//Li cells. In view of the aforementioned problem, anode-free cells are thus essential to disclosure the dead Li problem and for the respective evaluation of the battery performance.

Fig. R5. Results obtained from integrated protocol using 1M LiPF₆ in EC:DEC (1:1) as electrolyte under the current density of 0.2 mA cm⁻². **a**, normalized discharge capacity versus cycle number of NMC//Li and NMC//Cu cells. **b**, irreversible CE comparison of Li//Cu, NMC//Li, and NMC//Cu cells. The capacity retention comparison of NMC//Li and NMC//Cu cells are shown in Fig. S7a. (**Fig. 5** in the revised manuscript)

Comment #5: This work seems to emphasize on the fundamental understanding of Li behaviors, so the studied system cannot be limited in ester electrolyte system, but it should be extended to ether system to offer a general understanding in the both popular electrolytes. Based on this, it would be better to give some data in ether systems and extend the achieved conclusions and principles in ether-friendly battery systems.

Reply:

Thank you for the comment. In order to answer the reviewer's concern, we have performed the integrated protocol in the most common 1M LiTFSI in DME:DOL with 2 wt% of LiNO₃ electrolyte for the extension of our protocol in ether-based system and dissected the origins of irr-CEs in LFP//Cu cell accordingly.

Action:

(in the revised supporting information page 8-10.)

Extending the proposed protocol to ether system

We further extended our proposed protocol for evaluating the performance of AFLMB within the ether systems. 1M LiTFSI in DME:DOL (1:1) with 2 wt% of LiNO₃ added and lithium iron phosphate (LiFePO₄, LFP) are selected as the electrolyte and cathode material for demonstration, respectively. For the 1st cycle irr-CE of Li//Cu cell (See **Fig. S14b**), the total irr-CE of 1.91% can be separated into 0.31% of 1st extra SEI formation and 1.60% of dead Li + sub. SEI. Subsequently, the irr-CE at the 2nd cycle is 1.60% and keeps around ~1% in the subsequent cycles, which can be attributed to dead Li + sub. SEI. Meanwhile, LFP//Li cell shows the 1st cycle irr-CE of 2.93% for 1st cathode irreversible capacity. It is noted that the irr-CEs after the 1st cycles are mostly greater than 1%, which is higher than those in carbonate systems and can be explained as greater amount of oxidative electrolyte decomposition in ether-based system than that in carbonate-based electrolyte due to the lower upper limit of potential window for ethers. Moreover, the fraction of cathode degradation in the irr-CE in the subsequent cycles can be calculated from the normalized capacity retention, which is ~0.17% in each cycle (see the gray bar in **Fig. S14b**). Last but not the least, the 1st cycle irr-CE in LFP//Cu cell is 11.30%, which is much higher than that of 1.91% in Li//Cu cell and 2.93% in LFP//Li cell. However, it is confirmed that the capacity loss in the 1st cycle mainly originated from the loss of active Li inventory, since the 1st cycle lost capacity of LFP could be restored after re-assembling the cycled LFP electrode with metallic Li anode (see **Fig. S15a**). Based on the result, we suggest that the initial A/C ratio of LFP//Cu cell is less than one due to higher capacity loss at the anode and the irr-CE should be dominated by the anode. Thus, apart from the 0.31% of 1st extra SEI formation and 1.60% of dead Li + sub. SEI, the remained 9.39% of irr-CE is suggested to be cross-talk effects. In the subsequent cycles, the irr-CEs in LFP//Cu cell are still larger than those in Li//Cu cells, suggesting cross-talk effects hold a remarkable proportion of irr-CE in LFP//Cu cell when using 1M LiTFSI in DME:DOL (1:1) with 2 wt% LiNO₃ added as the electrolyte. Thus, the irr-CE in the 2nd cycle can be attributed to 1.6% dead Li + sub. SEI and 2.56% cross-talk effects, and that in the 20th cycle to 1.06% dead Li + sub. SEI and 1.01% cross-talk effects. In addition, we reassembled the cycled LFP electrode (from the dead LFP//Cu cell) with metallic Li anode to confirm the origin of capacity loss in LFP//Cu cell (See **Fig. S15b**). The results show that the reversible capacity of LFP//Li

cell is regained back to the level of the 1st charge capacity, thus, suggesting the capacity loss in LFP//Cu cell is mainly due to the irreversible reactions from anode, namely dead Li and SEI formation and the cross-talk effects. From the information obtained from the integrated protocol, it can be concluded that dead Li formation is significantly suppressed in ether-based electrolyte with the addition of LiNO₃, which is similar to the previous works reporting the stabilizing effect of LiNO₃ to metallic Li anode⁹. Thus, the capacity decay rate of the anode-free LFP//Cu cell is remarkably retarded compared to that of NMC//Cu cells in carbonate systems.

Fig. R8. Results obtained from the integrated protocol using 1M LiTFSI in DME:DOL (1:1) with 2 wt% of LiNO₃ added as electrolyte under the current density of 0.17 mA cm⁻² (0.1 c-Rate to the LFP electrode). **a**, Normalized discharge capacity versus cycle number of LFP//Li and LFP//Cu cells. **b**, irreversible CE comparison of Li//Cu, LFP//Li, and LFP//Cu cells at different cycles. (Fig. S14 in the revised supporting information, page 9)

Fig. R9. a, Charge/discharge curves of LFP//Cu in the 1st charge process and reassemble the LFP electrode with metallic Li anode for discharge process and the subsequent cycles. **b** areal capacity retention of LFP//Cu cell and the reassembled LFP//Li cell using the same LFP electrode in LFP//Cu cell. (Fig. S15 in the revised supporting information, page 10)

Comment #6: Partial descriptions should be concisely phrased and more scientific, e.g., “just like a rose always has its thorns”

Reply:

Thank you for the comment. The terms have been modified and highlighted in red.

Action:

“Although AFLMBs suffer from high irreversible coulombic efficiency and quick capacity fading and often recognized as a poor cell configuration, AFLMB could also serve as an indispensable key in facilitating the development of better electrolytes and evaluating the performance of LMBs.” (In the revised manuscript page 11)

Comment #7: Some related contributions can be referred to help the better express of this article. i) For the current strategies for Li metal anodes in Introduction part, recommend citing “Tuning wettability of molten lithium via a chemical strategy for lithium metal anodes, Nature Communications, 2019, 10, 1-8” and “A 3D Lithium/Carbon Fiber Anode with Sustained Electrolyte Contact for Solid-State Batteries, Advanced Energy Materials, 2019, 10 (3), 1903325.” ii) For the Li deposition/dissolution part, recommend

citing “Towards Better Li Metal Anodes: Challenges and Strategies, *Materials Today*, 2020, 33, 56-74.”

Reply:

Thank you for the comments. The above mentioned references have been included into the introduction part of the manuscript.

Action:

Reference number: (in the revised manuscript page 13 and 14)

11. Zhang Y, *et al.* Towards better Li metal anodes: Challenges and strategies. *Materials Today* **33**, 56-74 (2020).
19. Zhang Y, *et al.* A 3D Lithium/Carbon Fiber Anode with Sustained Electrolyte Contact for Solid-State Batteries. *Advanced Energy Materials* **10**, (2019).
20. Wang SH, *et al.* Tuning wettability of molten lithium via a chemical strategy for lithium metal anodes. *Nat Commun* **10**, 4930 (2019).

Reference for Reply:

1. Fang C, *et al.* Quantifying inactive lithium in lithium metal batteries. *Nature* **572**, 511-515 (2019).
2. Thirumalraj B, *et al.* Nucleation and Growth Mechanism of Lithium Metal Electroplating. *J Am Chem Soc* **141**, 18612-18623 (2019).
3. Kang S-H, Yoon W-S, Nam K-W, Yang X-Q, Abraham DP. Investigating the first-cycle irreversibility of lithium metal oxide cathodes for Li batteries. *Journal of Materials Science* **43**, 4701-4706 (2008).
4. Genovese M, Louli AJ, Weber R, Hames S, Dahn JR. Measuring the Coulombic Efficiency of Lithium Metal Cycling in Anode-Free Lithium Metal Batteries. *Journal of The Electrochemical Society* **165**, A3321-A3325 (2018).
5. Zhou H, Xin F, Pei B, Whittingham MS. What Limits the Capacity of Layered Oxide Cathodes in Lithium Batteries? *ACS Energy Letters* **4**, 1902-1906 (2019).
6. Burns JC, *et al.* Introducing Symmetric Li-Ion Cells as a Tool to Study Cell Degradation Mechanisms. *Journal of The Electrochemical Society* **158**, (2011).
7. Betz J, *et al.* Cross Talk between Transition Metal Cathode and Li Metal Anode: Unraveling Its Influence on the Deposition/Dissolution Behavior and Morphology of Lithium. *Advanced Energy Materials* **9**, (2019).

8. Zhang X-Q, *et al.* Crosstalk shielding of transition metal ions for long cycling lithium–metal batteries. *Journal of Materials Chemistry A* **8**, 4283-4289 (2020).
9. Li W, *et al.* The synergetic effect of lithium polysulfide and lithium nitrate to prevent lithium dendrite growth. *Nat Commun* **6**, 7436 (2015).

Reviewer #1 (Remarks to the Author):

I appreciate the efforts that the authors made to revise the manuscript. Several technical issues and concerns have been addressed and the manuscript has been improved. However, I still think that this paper is not suitable for Nature Communication considering the journal's broad impact. Evaluating Li//Cu, NMC//Li, and NMC//Cu cell data together provides more comprehensive information on the degradation mechanism of cathodes and lithium anodes. However, the suggested method of quantifying the precise contribution of each parameter is still weak in scientific rigor and physical basis. I doubt that this protocol can be used as a standard tool for cell-to-cell evaluation of different LMB and AFLMB cells because the parameters are "dissected" unclear and their quantification is based on assumptions. (parameters such as "1st irr-cap. of cathode", "ox E.D.", "sub SEI", "cross-talks", etc. are ambiguously defined without clear physical basis and mutually correlated.) Nevertheless, this paper may provide some useful hints and guidance when published in a battery specific journal.

Reviewer #2 (Remarks to the Author):

Our comments have been well-addressed. The revised manuscript is acceptable for publication.

The authors express their sincere appreciation to the editor and the reviewers for accepting our manuscript and providing thoughtful comments and constructive suggestions, which helped us to improve the quality of this manuscript. The point-to-point response to the reviewers' comments is as follows.

Point-to-Point Response (response in blue font)

Reviewer #1:

I appreciate the efforts that the authors made to revise the manuscript. Several technical issues and concerns have been addressed and the manuscript has been improved. However, I still think that this paper is not suitable for Nature Communication considering the journal's broad impact. Evaluating Li//Cu, NMC//Li, and NMC//Cu cell data together provides more comprehensive information on the degradation mechanism of cathodes and lithium anodes. However, the suggested method of quantifying the precise contribution of each parameter is still weak in scientific rigorousness and physical basis. I doubt that this protocol can be used as a standard tool for cell-to-cell evaluation of different LMB and AFLMB cells because the parameters are "dissected" unclear and their quantification is based on assumptions. (parameters such as "1st irr-cap. of cathode", "ox E.D.", "sub SEI", "cross-talks", etc. are ambiguously defined without clear physical basis and mutually correlated.) Nevertheless, this paper may provide some useful hints and guidance when published in a battery specific journal.

Reply:

Thank you for the insightful comments again. We have provided the validation results of our protocol using TGC measurements and decoupled data from different electrolyte systems as the reviewer suggested in the first revised manuscript. However, indeed there are still some sources of the irr-CE that could not be completely separated, such as the contribution of dead Li and sub. SEI unless an experimental approach like TGC measurement is conducted. As for cross-talk effects, further study on experimentally quantifying the contribution is currently under investigation. Nonetheless, in this work, we focus on the proposed protocol, which can extract the information from the electrochemical results of Li//Cu, cathode//Li, and cathode//Cu cells to decouple the

contribution of separable irr-CEs. With the demystified origins of different irr-CEs, one may comprehensively evaluate the effects of different approaches on improving the average CE of the LMB and AFLMB cells. Therefore, we believe that our protocol would be very helpful for scientists to develop anode-free lithium metal batteries or lithium metal batteries in various electrolytes. Meanwhile, the concept can be extended to other metal batteries like Na, K, Zn batteries etc and attract significant interests from the broad audience of Nature Communication.

Reviewer #2 :

Our comments have been well-addressed. The revised manuscript is acceptable for publication.

Reply:

Thank you again for your constructive comments, which enabled us to improve our work.